# TIME SERIES AS VIDEOS: SPECTRO-TEMPORAL GENERATIVE DIFFUSION

## ABSTRACT

Generative modeling of multivariate time series is challenged by properties such as non-stationarity, intricate cross-channel correlations, and multi-scale temporal dependencies. Existing diffusion models for this task mainly operate directly in the time-domain, employing architectures that are not designed to capture complex spectral dynamics. Conversely, methods that transform sequences into static images collapse the temporal axis, precluding the use of models designed for spatiotemporal dynamics. This paper argues for a new, unifying paradigm: reframing time series as videos. To this aim, we introduce Spectro-Temporal Diffusion (ST-Diff), a framework that leverages the Short-Time Fourier Transform (STFT) to convert a multivariate time series into a time-frequency video tensor. In this representation, frequency and covariate axes form the spatial dimensions of each frame, while the temporal evolution of the frequency spectrum is explicitly preserved. To capitalize on this novel structure, we design a custom video diffusion model specifically to leverage the spectro-temporal dynamics - the evolution of frequency components over time. Through extensive empirical evaluation on standard benchmarks, we demonstrate that the novel time-series-as-videos representation, together with its tailored architecture, allows ST-Diff to establish a new state-of-the-art in unconditional time series generation. We argue that this time-series-as-video paradigm has significant potential to advance a broad spectrum of sequence modeling tasks beyond unconditional time-series generation.

## 1 INTRODUCTION

Generative modeling of multivariate time series is a fundamental problem in machine learning with applications in financial simulation, climate forecasting, and privacy-preserving medical data Yoon et al. (2019); Esteban et al. (2017), among others. A core technical challenge is to generate synthetic samples that are statistically indistinguishable from real data, capturing not only the marginal distributions of variables but also their complex temporal dynamics. Real-world time-series are frequently characterized by properties such as non-stationarity, long-range dependencies, multi-scale periodicities, and aperiodic events, which makes their generation a particularly challenging task.

The recent success of diffusion models has driven a new wave of research in time series generation. A significant fraction of this work operates directly in the time domain, employing architectures like Recurrent Neural Networks (RNNs) or Transformers as the denoising backbone Rasul et al. (2021); Tashiro et al. (2021). While effective, RNN-based models often struggle to capture very long-range dependencies Bengio et al. (1994), while Transformer-based approaches, despite their power, may not possess the ideal inductive bias for modeling the nature of time series and can be computationally expensive for very long sequences Zeng et al. (2023).

An alternative line of work reframes the problem by transforming time series into static images, leveraging powerful computer vision architectures Wang & Oates (2015); Naiman et al. (2024). Techniques like delay embedding, Gramian angular fields and short-term Fourier transform map a sequence to a 2D matrix, enabling the use of state-of-the-art image diffusion models. This approach, however, collapses the temporal dimension into a spatial one. As a result, architectures designed to process spatiotemporal data cannot be used. This limitation motivates a key question: *Is it possible to design a time-series representation that reveals its internal frequency structure while preserving its native, explicit temporal axis, in order to leverage specialized spatiotemporal architectures?*

In this paper, we argue that the optimal representation for this task is not a static, 2D image, but a 3D video. Consequently, we introduce a new paradigm that treats time series generation as a task in the video domain. Our method uses the short-time Fourier transform (STFT), a central tool in signal processing, to convert a multivariate time series into an evolving time-frequency video tensor. In this representation, each frame is a matrix where one axis corresponds to frequency components and the other to the covariates. Crucially, the temporal evolution of the time-series frequency content is explicitly maintained along the video time axis. This transformation allows for the application of customized versions of *video diffusion models*, which are architecturally suited to learn how spatial patterns - in our case, frequency spectra in particular - evolve over time.

We present Spectro-Temporal Diffusion (ST-Diff), a generative diffusion framework that leverages this novel paradigm. The ST-Diff pipeline consists of three main steps: an invertible STFT-based mapping from the time series to a video tensor, a generative diffusion process on this video representation through a custom spectro-temporal model, and an inverse STFT to reconstruct the final time-domain signal. Our extensive experiments on public benchmarks show that this approach establishes a new state-of-the-art for unconditional time series generation, outperforming existing time-domain and image-based methods.

The contributions of this work are threefold:

1. We propose and formalize the treatment of time series generation as a video task, a method that preserves the temporal dimension while enabling the use of spatiotemporal models.

2. We introduce ST-Diff, a framework that integrates the STFT with a spectro-temporal video diffusion model to generate high-fidelity and dynamically consistent time series.

3. We empirically demonstrate that ST-Diff significantly outperforms prior state-of-the-art diffusion models on standard unconditional generation tasks.

We believe this *time-series-as-video* perspective offers a powerful and generalizable foundation that has the potential to advance a wide array of time series tasks, from forecasting to anomaly detection.

## 2 RELATED WORKS

Our work is situated at the intersection of generative models for time series, time series data representation, and video diffusion models. We review key developments in these areas to contextualize our contributions.

**Generative Models for Time Series** Prior to diffusion models, generative modeling for time series was primarily advanced by Generative Adversarial Networks (GANs) and Variational Autoencoders (VAEs). GAN-based models such as RCGAN Esteban et al. (2017) and TimeGAN Yoon et al. (2019) employ an adversarial training objective, with TimeGAN notably incorporating a supervised loss to better capture temporal correlations. VAEs, including models like TimeVAE Desai et al. (2021), offer a stable, likelihood-based alternative and can incorporate interpretable latent spaces. Our work leverages diffusion models, which have demonstrated superior sample fidelity and training stability compared to these earlier approaches Yuan & Qiao (2024).

The application of Denoising Diffusion Probabilistic Models (DDPMs) to time series has largely focused on models that operate directly on the raw signal. Initial works such as TimeGrad Rasul et al. (2021) for forecasting and CSDI Tashiro et al. (2021) for imputation adapted the diffusion process for conditional tasks, typically using RNN or Transformer-based networks for the denoising step. For the category of unconditional generation, Diffusion-TS Yuan & Qiao (2024) represents a milestone, and employs a decomposition architecture to explicitly model trend and seasonality components. While Diffusion-TS uses a Fourier-based loss to enforce periodicity, its core diffusion process remains in the time domain. This contrasts with our approach, where the time-frequency representation is not a supervisory signal but the primary domain for the entire generative process. Complementary to this line of work, Crabbé et al. (2024) propose frequency diffusion models that perform the entire generative process in the frequency domain, whereas our approach operates directly in the joint time–frequency plane, capturing temporal and spectral structures simultaneously.

**Time Series to Image Transformations**  A parallel research direction involves transforming time series into 2D image representations to leverage well established, powerful vision architectures. This concept was explored using methods like gramian angular fields and recurrence plots Wang & Oates (2015). The leading contemporary model in this paradigm is ImagenTime Naiman et al. (2024), which uses invertible transforms such as delay embedding and STFT to encode a time series into a single, 2D static image. A standard vision diffusion model is then trained on these images. While this approach has proven highly effective, it effectively treats the the temporal axis as a spatial one. The explicit temporal sequence is lost, precluding the use of architectures designed for spatiotemporal modeling. Our work addresses this limitation directly by proposing a video representation that reveals the time series frequency structure without sacrificing its temporal dimension.

**Time-Frequency Representations and Video Generation**  The Short-Time Fourier Transform (STFT) is a central method in signal processing for obtaining a time-frequency representation, revealing the temporal evolution of a signal spectral content Allen & Rabiner (1977). This representation, visualized as a spectrogram, is foundational in audio generation Shen et al. (2018). Concurrently, video generation has seen rapid progress, with video diffusion models demonstrating the ability to synthesize high-fidelity, temporally coherent video sequences Ho et al. (2022). To our knowledge, our work is the first to systematically bridge these domains for general multivariate time series generation. We argue that the video tensor derived from the STFT of a multivariate time series is a more natural and informative representation than either the raw signal or a static image. Unlike Diffusion-TS, we model the dynamics of the spectrum itself. Unlike ImagenTime, we preserve the temporal axis explicitly, obtaining a spatiotemporal representation that enables ST-Diff to use spatiotemporal models to learn the evolution of a time series frequency components.

## 3  BACKGROUND

Our framework integrates two core techniques: the STFT for data representation and DDPMs adapted for video generative diffusion. We briefly review these concepts and establish the notation used throughout this paper.

**Short-Time Fourier Transform (STFT)**  The STFT maps a time-domain signal to a time-frequency representation describing the temporal evolution of its frequency content. Given a one-dimensional discrete-time signal $x[n]$ of length $L$, its discrete STFT, $X[m,k]$, is a complex-valued matrix computed as: $X[m,k] = \sum_{n=0}^{L-1} x[n]w[n-mH]e^{-j\frac{2\pi kn}{N}}$, where $w[\cdot]$ is a window function $w[\cdot]$ which can mitigate spectral leakage (e.g., Hann window), $m$ is the time frame index, and $k$ is the discrete frequency index. The STFT is controlled by two main hyperparameters: the window length $N$, which determines the trade-off between time and frequency resolution (resulting in an uncertainty principle); the hop length $H$ (step size between the start of consecutive windows), which controls the temporal resolution of the representation. A critical property for our generative framework is the invertibility of the STFT. The original signal $x[n]$ can be reconstructed from $X[m,k]$ via the inverse STFT (iSTFT), typically using an overlap-add synthesis method. This near-perfect reconstruction ensures that samples generated in the time-frequency domain can be losslessly converted back to the time domain.

**Video Diffusion Models**  DDPMs are a class of generative models that learn to reverse a fixed noise-injection process, and can be adapted for video data. Let $V_0 \in \mathbb{R}^{T \times C \times H \times W}$ be a clean video tensor, where $T$ is the number of frames. The forward process, $q$, is a fixed Markov chain that gradually adds Gaussian noise to the data over $T_{\text{diff}}$ discrete timesteps: $q(V_t|V_{t-1}) = \mathcal{N}(V_t; \sqrt{1-\beta_t}V_{t-1}, \beta_t\mathbf{I})$ where $\{\beta_t\}_{t=1}^{T_{\text{diff}}}$ is a predefined variance schedule. It is possible to sample $V_t$ at an arbitrary timestep $t$ in closed form: $V_t = \sqrt{\bar{\alpha}_t}V_0 + \sqrt{1-\bar{\alpha}_t}\epsilon$, where $\alpha_t = 1 - \beta_t$, $\bar{\alpha}_t = \prod_{i=1}^{t}\alpha_i$, and $\epsilon \sim \mathcal{N}(0,\mathbf{I})$. The generative model, $p_\theta$, learns to approximate the reverse process $p(V_{t-1}|V_t)$. This is achieved by training a neural network $\epsilon_\theta(V_t, t)$ to predict the noise component $\epsilon$ from the noisy input $V_t$ at timestep $t$. The network is optimized with a mean-squared error loss on the noise: $\mathcal{L} = \mathbb{E}_{t,V_0,\epsilon}\left[||\epsilon - \epsilon_\theta(\sqrt{\bar{\alpha}_t}V_0 + \sqrt{1-\bar{\alpha}_t}\epsilon, t)||^2\right]$. For video data, the network $\epsilon_\theta$ is typically implemented as a spatiotemporal architecture, which model dependencies both within and across video frames. Generation of a new video is performed by starting with a random noise tensor $V_{T_{\text{diff}}} \sim \mathcal{N}(0,\mathbf{I})$ and iteratively applying the learned denoising function to sample $V_{t-1}$ from

$V_t$, until a clean sample $V_0$ is produced. In our work, we apply this generative mechanism not to natural videos of scenes, but to the time-frequency videos derived from time series data.

# 4 THE SPECTRO-TEMPORAL DIFFUSION (ST-DIFF) FRAMEWORK

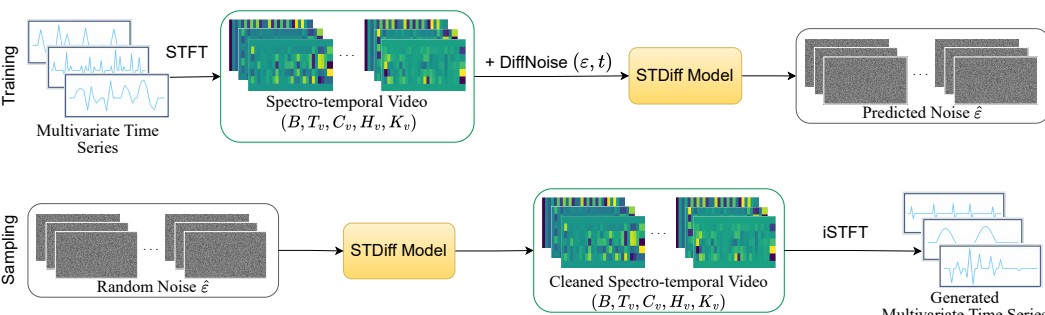

Figure 1: Overview of the Spectro-Temporal Diffusion (ST-Diff) pipeline. For training (top), a multivariate time series is transformed into a spectro-temporal video tensor via trend-residual decomposition and the STFT. For sampling (bottom), the learned STDiff model generates a new tensor in this domain, which is then converted back to a time series using the inverse STFT (iSTFT).

We introduce Spectro-Temporal Diffusion (ST-Diff), our proposed framework for multivariate time series generation. The core of our method is to first transform the time series into a spectro-temporal video representation via the STFT transformation and then apply a specialized video diffusion model to generate samples directly in this domain. These are subsequently converted back to the time domain using the inverse STFT (iSTFT). An overview of the full pipeline is illustrated in Figure 1.

## 4.1 FROM TIME SERIES TO SPECTRO-TEMPORAL VIDEO TENSORS

A multivariate time series is a tensor $X \in \mathbb{R}^{L \times K}$, where $L$ is the sequence length and $K$ is the number of covariates. Our transformation pipeline maps $X$ to a video tensor $V \in \mathbb{R}^{T \times C \times H \times W}$. As shown in Fig. 1 and Fig. 2a, we start by decomposing each covariate channel $\boldsymbol{x}_k \in \mathbb{R}^L$ into a trend component $\boldsymbol{x}_{k,\text{trend}}$ and a residual component $\boldsymbol{x}_{k,\text{res}}$, in order to handle non-stationarity taht is common in real-world time series. We compute the trend using a simple exponential moving average (EMA). This isolates the low-frequency, non-stationary behavior, leaving the residual component, $\boldsymbol{x}_{k,\text{res}} = \boldsymbol{x}_k - \boldsymbol{x}_{k,\text{trend}}$, which is more suitable for spectral analysis, as the STFT is most effective on quasi-stationary signals.

Then, we apply the STFT independently to each of the $K$ residual sequences, $\boldsymbol{x}_{k,\text{res}}$. This produces $K$ complex-valued time-frequency matrices, $\{\boldsymbol{S}_k \in \mathbb{C}^{F \times T}\}_{k=1}^K$, where $F$ is the number of frequency bins and $T$ is the number of time frames, determined by the STFT hyperparameters (window size $N$ and hop length $H$). To form a real-valued tensor suitable for neural network processing, we construct the final video tensor $V$, whose dimensions are: the temporal axis of the video corresponds to the STFT time frames, $T$; The height of each frame corresponds to the frequency bins, $F$; The width of each frame corresponds to the covariates, $K$; Three channels ($C = 3$, with the first two storing the real and imaginary parts of the STFT coefficients, $\text{Re}(\boldsymbol{S}_k)$ and $\text{Im}(\boldsymbol{S}_k)$, while the third channel stores the trend component, $\boldsymbol{x}_{k,\text{trend}}$, which is broadcasted across the frequency dimension and resampled to match the temporal dimension $T$). This process yields a final tensor $V \in \mathbb{R}^{T \times 3 \times F \times K}$. This representation explicitly preserves the temporal evolution of the signal spectral content across all covariates, making it directly compatible with video generation models.

## 4.2 GENERATION AND INVERSE TRANSFORMATION

To generate a new time series, we first sample a noise tensor $V_{T_{\text{diff}}} \sim \mathcal{N}(0, \mathbf{I})$ and apply the reverse diffusion process using the trained model $\epsilon_\theta$ to obtain a synthetic spectro-temporal video tensor $V_{\text{gen}}$ (see Fig. 1). This tensor is then inverted back to the time domain. The three channels of $V_{\text{gen}}$ are

separated to recover the real and imaginary parts of the STFT for each covariate, as well as the trend components. The inverse STFT (iSTFT) is applied to the generated spectogram of each covariate to reconstruct the residual signals, $\hat{x}_{k,\text{res}}$. Adding the generated trend back to the residual yields the final time series for each covariate: $\hat{x}_k = \hat{x}_{k,\text{res}} + \hat{x}_{k,\text{trend}}$. This process yields the final synthetic multivariate time series $\boldsymbol{X}_{\text{gen}} \in \mathbb{R}^{L \times K}$.

## 4.3 THE SPECTRO-TEMPORAL VIDEO DIFFUSION MODEL

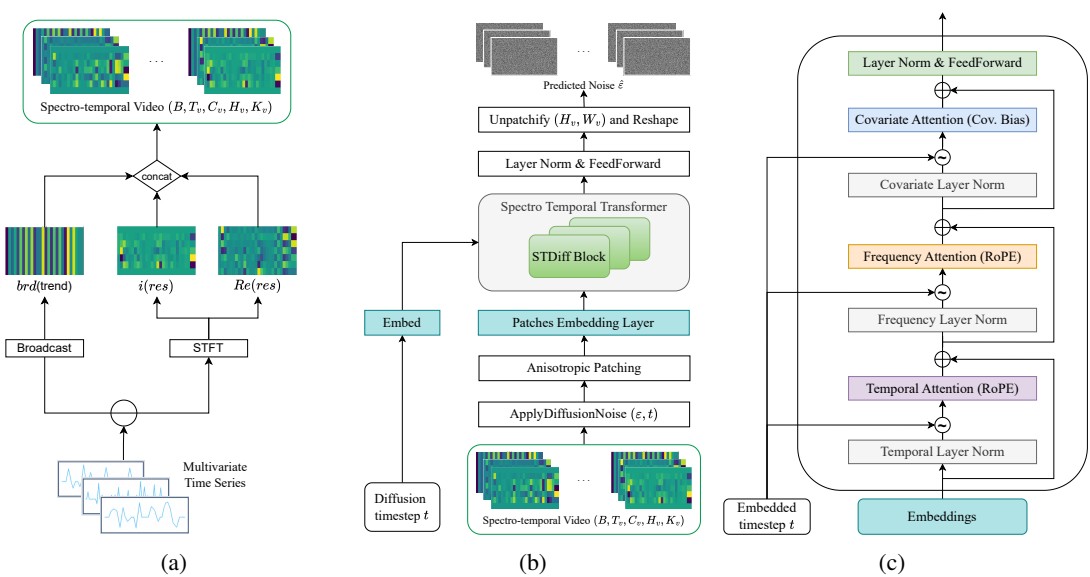

(a)      (b)      (c)

Figure 2: Overview of the proposed STDiff data pipeline and core model architecture. Panel 2a: The data ingestion pipeline, transforming a multivariate time series into a spectro-temporal video tensor. Panel 2b: The main STDiff model. Panel 2c: The Spectro-temporal attention block, showing the tri-axial factorized attention mechanism.

With the data represented as a spectro-temporal video tensor, we employ a DDPM $\epsilon_\theta$ for generation. The architecture of $\epsilon_\theta$ is a key component of our framework, which factorizes attention across the spatial, temporal, and covariate axes, with specific architectural biases for each of them. We outline the key architectural components below.

**Anisotropic Patching and Spectro-Temporal Attention Biases** The input tensor frame, a frequency–covariate matrix of shape $(F \times K)$, is first projected into a sequence of tokens. Unlike vision transformers that employ isotropic patches (e.g., $16 \times 16$), we adopt an *anisotropic* patching strategy: patches are aggregated along the frequency axis while preserving unit granularity along the covariate axis, so as not to introduce arbitrary spatial correlations among covariates, which, unlike in image data, we do not assume a priori.

The network backbone comprises a stack of *STDiff* blocks (Fig. 2c), which apply attention sequentially along the three main temporal, frequency, and covariate axes. To encode domain-specific structure, we introduce two bias mechanisms. First, the covariate attention module incorporates a symmetric matrix $\boldsymbol{B}_C \in \mathbb{R}^{K \times K}$ into its attention logits, yielding attention scores softmax$(\frac{\boldsymbol{QK}^T}{\sqrt{d_k}} + \boldsymbol{B}_C)\boldsymbol{V}$. This bias acts as a learnable prior over inter-covariate dependencies. Second, a frequency bias matrix $\boldsymbol{B}_F \in \mathbb{R}^{F' \times F'}$ (where $F'$ denotes the number of frequency patches) is analogously added to the frequency attention module, enabling the capture of structured relationships among spectral bands. Both bias matrices are initialized from empirical statistics of the data. Specifically, $\boldsymbol{B}_C$ is set to the empirical cross-correlation matrix of the STFT covariates, encoding static inter-variable dependencies intrinsic to the system. In parallel, $\boldsymbol{B}_F$ is initialized from the covariance of STFT log-magnitudes, thereby modeling spectral components that tend to co-vary (e.g., fundamental frequencies and harmonics). Our biases encourage the model to respect domain-

relevant structural and spectral relationships (with a role akin to spatial locality in convolutions). Crucially, this is well-aligned with the underlying data: the covariate axis represents an *unordered set* of variables with no notion of locality, while spectral dependencies are often highly non-local.

**Positional and Timestep Embeddings**  We use Rotary Positional Embeddings (RoPE) Su et al. (2024) to encode the relative positions of tokens along the temporal and frequency axes, as they are suitable to capture the relative ordering without being constrained to a fixed maximum length. The covariate positions, instead, are encoded using a learnable parameters vectors, due to inherently non-ordered structure of the covariate axis. Standard sinusoidal embeddings are used to encode the diffusion timestep $t$, which are then processed by a multi-layer perceptron (MLP) before being incorporated into the network blocks. The timestep embedding is integrated into the transformer blocks using an adaptive layer normalization scheme (adaLN-Zero) Peebles & Xie (2023).

## 5 EXPERIMENTS

We conduct a comprehensive set of experiments to evaluate the performance of ST-Diff for unconditional multivariate time series generation. Our evaluation is designed to assess distributional fidelity, sample quality, and the preservation of temporal dynamics.

**Datasets**  We evaluate our method on six publicly available benchmark datasets spanning diverse properties such as dimensionality, periodicity, and non-stationarity, consistent with prior work (Naiman et al., 2024; Yuan & Qiao, 2024). The datasets are: Sines, synthetic sine waves with varying frequencies and phases (a sanity check to test the model ability to capture fundamental periodic patterns); Stocks, daily stock prices exhibiting non-stationary stochastic behavior; ETTh, electricity transformer temperature real-world data with strong periodic components; Energy, appliance energy consumption with multivariate correlations and noisy periodicity; MuJoCo, high-dimensional physics simulator data capturing complex dynamics; and fMRI, high-dimensional neural signals characterized by noise and correlations. Following standard evaluation protocols, all datasets Following standard evaluation protocols, we use a sequence length of $L = 24$ across all datasets. To further assess model scalability, we additionally evaluate on the ETTh dataset with longer sequence lengths of $L \in 64, 128, 256$.

**Evaluation Metrics**  To assess generation quality, we use an established suite of quantitative and qualitative metrics Yoon et al. (2019), all reported so that lower values indicate better performance. The Discriminative Score is measured by training a GRU classifier to distinguish real from synthetic data. The score is the absolute difference between the classifier accuracy on a held-out test set and 0.5 (random chance). A score near zero indicates that the generated samples are indistinguishable from real ones. The Predictive Score evaluates the usefulness and the preservation of temporal dynamics through the "Train on Synthetic, Test on Real" protocol, where a GRU one-step-ahead forecaster trained on generated data is tested on real data and its Mean Absolute Error (MAE) is reported. To capture cross-covariate structure, we report the Correlational Score, computed as the mean absolute difference of the Pearson correlation matrices of the real and the generated dataset. Finally, we include qualitative analyses, such as t-SNE projections and data density estimations to compare distributional similarity, and comparisons of Auto-Correlation Function (ACF) and Power Spectral Density (PSD) to evaluate temporal and spectral fidelity.

**Baselines**  We compare ST-Diff against leading models and frameworks for time series generation: TimeGAN (Yoon et al., 2019), a GAN-based framework for sequential data; TimeVAE (Desai et al., 2021), a VAE-based generative model; Diffusion-TS (Yuan & Qiao, 2024), a state-of-the-art diffusion model operating directly in the time domain; and ImagenTime (Naiman et al., 2024), a diffusion-based approach that maps time series into images. For all baselines, we report performance from the original publications to ensure fair comparison.

**Implementation Details**  ST-Diff is implemented in PyTorch. The denoising network $\epsilon_\theta$ corresponds to the spectro–temporal video diffusion transformer introduced in Sec. 4.3. To construct the input representation the FFT size is scaled relative to the input duration as $\mathtt{nfft} = (\mathtt{seq\_len}/2) - 1$ with hop length set proportionally as $\lceil \mathtt{nfft}/4 \rceil$. This normalization transforms variable-length time

Table 1: Comprehensive quantitative comparison for unconditional generation on standard short sequences (L=24). We report Context-Fid, Correlational, Discriminative and Predictive scores ('lower is better'). **ST-Diff** sets a new state-of-the-art across the majority of metrics and datasets. The '–' symbol indicates that the metric was not reported in the original paper.

| Metric | Methods | Sines | Stocks | ETTh | MuJoCo | Energy | fMRI |
|---|---|---|---|---|---|---|---|
| Context-FID Score (Lower the Better) | TimeGAN | 0.101±.014 | 0.103±.013 | 0.300±.013 | 0.563±.052 | 0.767±.103 | 1.292±.218 |
| | TimeVAE | 0.307±.060 | 0.215±.035 | 0.805±.186 | 0.251±.015 | 1.631±.142 | 14.449±.969 |
| | ImagenTime | – | – | – | – | – | – |
| | DiffusionTs | 0.006±.000 | 0.147±.025 | 0.116±.010 | 0.013±.001 | 0.089±.024 | 0.105±.006 |
| | STDiff (ours) | **0.004±.001** | **0.040±.008** | **0.050±.008** | **0.010±.001** | **0.025±.002** | **0.099±.007** |
| Correlational Score (Lower the Better) | TimeGAN | 0.045±.010 | 0.063±.005 | 0.210±.006 | 0.886±.039 | 4.010±.104 | 23.502±.039 |
| | TimeVAE | 0.131±.010 | 0.095±.008 | 0.111±.020 | 0.388±.041 | 1.688±.226 | 17.296±.526 |
| | ImagenTime | – | – | – | – | – | – |
| | DiffusionTs | **0.015±.004** | 0.004±.001 | 0.049±.008 | **0.193±.027** | 0.856±.147 | **1.411±.042** |
| | STDiff (ours) | 0.015±.005 | **0.003±.003** | **0.047±.006** | 0.199±.017 | **0.592±.013** | 1.661±.059 |
| Discriminative Score (Lower the Better) | TimeGAN | 0.011±.008 | 0.102±.021 | 0.114±.055 | 0.238±.068 | 0.236±.012 | 0.484±.042 |
| | TimeVAE | 0.041±.044 | 0.145±.120 | 0.209±.058 | 0.230±.102 | 0.499±.000 | 0.476±.044 |
| | ImagenTime | – | 0.037±.006 | – | 0.007±.005 | 0.040±.004 | – |
| | DiffusionTs | 0.006±.007 | 0.067±.015 | 0.061±.009 | 0.008±.002 | 0.122±.003 | 0.167±.023 |
| | STDiff (ours) | **0.004±.005** | **0.015±.021** | **0.005±.005** | **0.007±.005** | **0.009±.013** | **0.021±.014** |
| Predictive Score (Lower the Better) | TimeGAN | 0.093±.019 | 0.038±.001 | 0.124±.001 | 0.025±.003 | 0.273±.004 | 0.126±.002 |
| | TimeVAE | 0.093±.000 | 0.039±.000 | 0.126±.004 | 0.012±.002 | 0.292±.000 | 0.113±.000 |
| | ImagenTime | – | 0.036±.000 | – | 0.033±.001 | 0.250±.000 | – |
| | DiffusionTs | **0.093±.000** | 0.036±.000 | 0.119±.002 | **0.007±.000** | 0.250±.000 | 0.099±.000 |
| | STDiff (ours) | 0.186±.004 | **0.033±.000** | **0.119±.002** | **0.007±.000** | **0.211±.000** | **0.077±.000** |

series into fixed-dimensional spectrograms, ensuring that the subsequent analysis is independent of the original sequence length. A 75% overlap between analysis windows is employed, consistent with the theoretical requirements for robust signal invertibility Griffin & Lim (1984).

The model is trained under the DDPM framework with $T_{\text{diff}} = 1000$ diffusion steps and a cosine noise schedule. The training objective is the mean-squared error between the true and predicted noise, as detailed in Sec. 3. To further improve the fidelity of generated samples, particularly in capturing spectral characteristics critical to time-series data, we introduce a cross-covariance loss applied directly to the Short-Time Fourier Transform (STFT) magnitudes. This loss quantifies the discrepancy between normalized covariance matrices, thereby encouraging the covariance structure of generated STFT magnitudes to align closely with that of the real data. Optimization is performed using AdamW with a cosine annehaling scheduler for the learning rate, with a minimum learnining rate of $1 \times 10^{-6}$ and a maximum learning rate of $2 \times 10^{-4}$. The maximum number of epochs is 1000, but an early stopping mechanism has been implemented. For sample generation, we employ the DDIM sampler (Song et al., 2022) with 200 steps, which accelerates inference while maintaining sample fidelity. All experiments are conducted on a single NVIDIA A100 GPU.

### 5.1 EMPIRICAL RESULTS AND ANALYSIS

We present the empirical evaluation of ST-Diff, beginning with a quantitative comparison against state-of-the-art baselines on standard benchmarks. We further investigate the scalability to longer sequence lengths and complement these results with qualitative analyses of the generated samples.

#### 5.1.1 SHORT-TERM UNCONDITIONAL GENERATION

Table 1 reports results for unconditional generation on sequences of length 24. We evaluate ST-Diff against all baselines using four established metrics: Discriminative, Predictive, Correlational and Context-FID scores, where lower values indicate better performance.

Across the majority of datasets and evaluation metrics, ST-Diff establishes a new state of the art, achieving superior performance on 21 out of 24 metric–dataset combinations. The improvements are especially pronounced on high-dimensional, real-world datasets such as ENERGY, MUJOCO, and FMRI. On ENERGY and FMRI benchmarks in particular, ST-Diff delivers substantial reductions in discriminative and predictive scores, highlighting its capacity to model intricate cross-channel dependencies and non-trivial spectral evolutions, generating high-fidelity samples. Taken together, the results provide strong empirical evidence that explicitly modeling spectro–temporal structure constitutes a powerful inductive bias for complex multivariate time series.

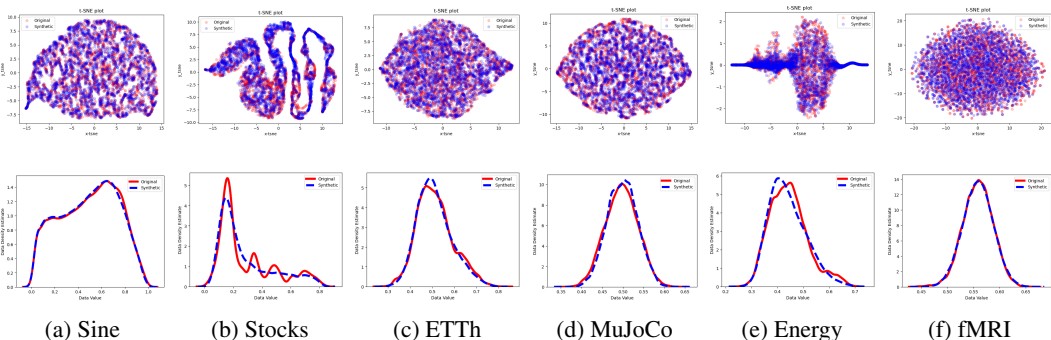

Figure 3: t-SNE (up), Kernel Density Estimation (bottom) of real vs generated samples of the 6 used datasets for sequence length 24. In red the original time series, in blue the generated ones.

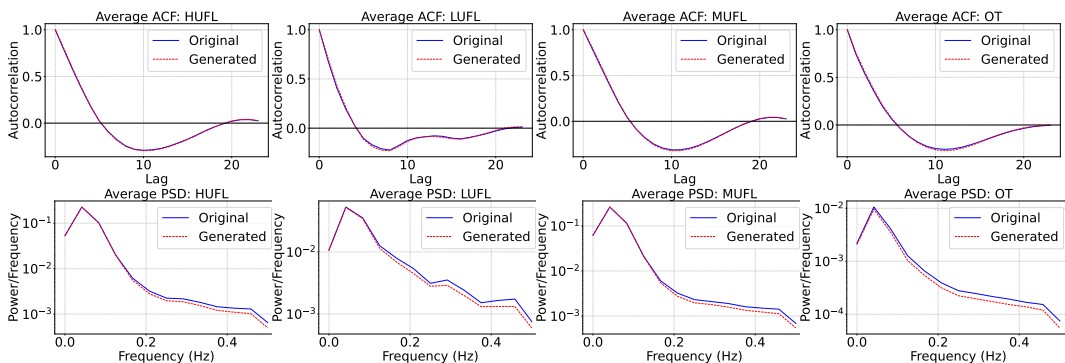

Figure 4: Temporal and spectral fidelity analysis on the ETTH dataset with sequence length 24. The top row reports the Auto–Correlation Function (ACF), and the bottom row the average Power Spectral Density (PSD), for real (blue, solid) and generated (red, dashed) samples across three representative covariates (HUFL, LUFL, MUFL, OT). The near-perfect overlap in the ACF plots and the close alignment of the PSD curves (in particular at low frequencies) demonstrate that ST-Diff faithfully reproduces both the temporal dependencies and the spectral characteristics of the original process. Extended ACF and PSD results are provided in Appendix C.

**Qualitative Analysis**   To complement the quantitative results, we provide qualitative visualizations. Figure 3 illustrates t-SNE embeddings and Kernel Density Estimation (KDE) of real and generated samples from all the datasets. The distribution of samples generated by ST-Diff closely aligns with the manifold of the real data. In the top row, the t-SNE projections offer a low-dimensional view of the high-dimensional time series, allowing a direct comparison between real (red) and synthetic (blue) distributions. Across all six datasets, the KDE curves of the generated samples (bottom row) closely follow those of the real data showing the alignment of marginal distributions and further evidence of the high generated sample fidelity achieved by ST-Diff.

To qualitatively assess directly temporal and spectral fidelity, we report a comparison of the average Auto–Correlation Function (ACF) and Power Spectral Density (PSD) of real and generated samples from the ETTH dataset (Fig. 4). The ACF plots (top row) show that ST-Diff accurately reproduces the temporal structure of the original series, indicating that it learns underlying dynamics rather than merely matching marginals. In the frequency domain, the PSD plots (bottom row) overall captures both dominant peaks and spectral decay, in particular at low-frequency components, with some slight difference in particular on high-frequency ones. Further results are reported in Appendix C.

### 5.1.2 LONG-TERM UNCONDITIONAL GENERATION

To assess the scalability of our approach, we evaluate performance on the ETTh datasets with extended sequence lengths of 64, 128, and 256, as summarized in Table 2. The findings unequivocally demonstrate the superior scalability of ST-Diff, which not only outperforms all baselines across

Table 2: Detailed results on long-term time-series generation for **ETTh**. (Lower is better.)

| Metric | Length | DiffusionTs | TimeGAN | TimeVAE | STDiff (ours) |
|---|---|---|---|---|---|
| Context-FID Score | 64 | 0.631±.058 | 1.130±.102 | 0.827±.146 | **0.031±.010** |
| | 128 | 0.787±.062 | 1.553±.169 | 1.062±.134 | **0.471±.003** |
| | 256 | 0.423±.038 | 5.872±.208 | 0.826±.093 | **0.341±.045** |
| Correlational Score | 64 | 0.082±.005 | 0.483±.019 | 0.067±.006 | **0.055±.015** |
| | 128 | 0.088±.005 | 0.188±.006 | 0.054±.007 | **0.036±.009** |
| | 256 | 0.064±.007 | 0.522±.013 | 0.046±.007 | **0.044±.019** |
| Discriminative Score | 64 | 0.106±.048 | 0.227±.078 | 0.171±.142 | **0.030±.020** |
| | 128 | 0.144±.060 | 0.188±.074 | 0.154±.087 | **0.032±.021** |
| | 256 | 0.060±.030 | 0.442±.056 | 0.178±.076 | **0.029±.042** |
| Predictive Score | 64 | 0.116±.000 | 0.132±.008 | 0.118±.004 | **0.071±.000** |
| | 128 | 0.110±.003 | 0.153±.014 | 0.113±.005 | **0.065±.000** |
| | 256 | 0.341±.045 | 0.220±.008 | 0.110±.027 | **0.074±.001** |

every metric and sequence length but often does so by a substantial margin. The advantage is particularly striking in the Context-FID score, where at a length of 64, ST-Diff achieves a score of 0.031, representing more than an order-of-magnitude improvement over the next-best competitor. This indicates a far more accurate and comprehensive approximation of the true data distribution's manifold. Furthermore, the degradation in ST-Diff is notably less pronounced as sequence length increases. While competing models show considerable performance degradation, ST-Diff's Discriminative Score remains exceptionally low and stable across all tested lengths ($0.030 \rightarrow 0.032 \rightarrow 0.029$). This suggests that the generated samples remain indistinguishable from real data even at longer horizons, a critical marker of a robust and well-generalized generative process. The model's capacity to preserve meaningful temporal dynamics is confirmed by its consistently superior Predictive Scores. It indicates that the fundamental, step-by-step transition dynamics learned from ST-Diff's synthetic data are faithful to the real process. These findings provide compelling evidence that our time-series-as-video paradigm is not only effective but overcomes a key limitation of models that operate purely in the time domain, which struggle with long contexts, or those that collapse the temporal axis into a static image representation, thereby losing explicit sequential information.

## 6 CONCLUSION

In this paper, we addressed a central challenge in generative modeling of multivariate time series: balancing expressive representations with faithful preservation of temporal structure. Existing approaches either operate directly in the time domain, limiting their ability to capture spectral properties, or transform sequences into static images, collapsing the temporal axis and precluding spatiotemporal modeling.

To solve these limitations, we introduced *Spectro-Temporal Diffusion* (ST-Diff), which reframes time series as videos for generative diffusion. ST-Diff maps a multivariate time series to a spectro-temporal video tensor via the short-time Fourier transform (STFT), explicitly preserving the evolution of spectral content over time and making the problem amenable to modern video diffusion architectures. We further developed a specialized spatiotemporal transformer with inductive biases tailored to this domain, enabling effective learning of complex spectro–temporal dynamics.

Our extensive empirical study demonstrates that ST-Diff establishes a new state of the art in unconditional time series generation, consistently outperforming time-domain and image-based diffusion models across diverse benchmarks, with particularly strong gains on high-dimensional, complex datasets. These findings suggest that unifying classical signal-processing principles with spatiotemporal generative modeling through our *time-series-as-video* approach yields a powerful and generalizable foundation for sequence generation.

Despite its strong performance, ST-Diff incurs higher computational and memory costs than time- or image-based models due to the use of spatiotemporal architectures. Exploring more efficient video-generation paradigms, such as latent video diffusion or model distillation, may mitigate this overhead. The proposed paradigm also opens several avenues for future research: extending ST-Diff to conditional tasks (e.g., forecasting and imputation), leveraging learned spectral-temporal distributions for unsupervised anomaly detection, and applying the approach to other sequential data domains where time–frequency analysis is essential, including audio, EEG, and seismic signals.

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

# A   DATASETS AND METRICS

## A.1   DATASETS

Our evaluation uses six publicly available datasets, chosen to span a wide range of characteristics including synthetic and real-world data, varying sequence lengths, and different levels of dimensionality and non-stationarity. This selection is consistent with prior work in time series generation [1, ImagenTime, Diffusion-TS].

- Sines: A synthetic dataset of sine waves with varying frequencies and phases, used to test a model ability to learn fundamental periodic patterns.
- Stocks: Real-world daily stock price data (Google), characterized by non-stationary behavior and random walks.
- ETTh: Electricity Transformer Temperature data, containing high-frequency, multivariate measurements with strong periodicities.
- MuJoCo: High-dimensional data from a physics simulator, representing complex and non-linear dynamics.
- Energy: Real-world appliance energy consumption data, featuring multivariate correlations and noisy periodicity.
- fMRI: High-dimensional functional magnetic resonance imaging data, characterized by noisy, complex, and correlated signals.

| Dataset | # Samples | # Covariates |
|---------|-----------|--------------|
| Sines   | 10,000    | 5            |
| Stocks  | 3,773     | 6            |
| ETTh    | 17,420    | 7            |
| MuJoCo  | 10,000    | 14           |
| Energy  | 19,711    | 28           |
| fMRI    | 10,000    | 50           |

Table 3: Overview of datasets used, including number of samples and covariates.

## A.2 EVALUATION METRICS

To provide a comprehensive and robust assessment of our model performance, we evaluate the quality of the generated time series using a suite of four distinct, literature-established metrics Yoon et al. (2019). These metrics are designed to measure from low-level statistical properties to high-level temporal dynamics. All metrics are designed to be "the-lower-the-better."

- Context-FID: To assess the overall distributional similarity between the real and synthetic datasets, we employ the Fréchet Inception Distance adapted for time series (Context-FID). We first use a pre-trained TS2Vec model to generate a single, holistic embedding for each time series in both the real and generated sets. The FID score is then calculated between these two distributions of embeddings. A low Context-FID indicates that the model is successfully capturing the diversity and global characteristics of the true data distribution.

- Cross-Correlation: We measure the model ability to preserve the complex inter-feature relationships using a cross-correlation metric. This metric computes the cross-correlation matrix between all pairs of co-variates for both the real and generated data. The final score is the aggregate difference between these two matrices. A low score signifies that the model is correctly learning and reproducing the instantaneous structural dependencies between the different time series co-variates.

- Discriminative Score: To evaluate the sample-level realism of the generated data, we use an adversarial approach. A separate, post-hoc GRU-based classifier is trained from scratch with the task of distinguishing between real and synthetic time series. The final Discriminative Score is the absolute difference between the classifier accuracy and 0.5 (random chance). A score close to zero indicates that the generated samples are of high fidelity and are indistinguishable from real data.

- Predictive Score: To evaluate if the generated data preserves the underlying temporal dynamics of the original series, we employ a "Train-on-Synthetic, Test-on-Real" (TSTR) evaluation. A simple GRU-based forecasting model is trained exclusively on the synthetic data to predict one step ahead. This trained model is then tested on the real, unseen data. The reported Predictive Score is the Mean Absolute Error (MAE) of these predictions. A low score demonstrates that the temporal patterns learned from the synthetic data are meaningful and can generalize to the real-world dynamics.

## B MODEL ARCHITECTURAL PARAMETERS

In Table 4 we provide a detailed description of the configuration hyperparameters of the STDiff model in the experiments presented in the sections above. A noteworthy observation is the consistent parametrization of Hidden Size and Num Heads for sequence lengths 64, 128, and 256. This choice reflects an architectural stability, maintaining a robust representational capacity and multi-head attention mechanism across varying input durations that likely demand similar levels of feature complexity and contextual integration. However, the configuration for sequence length 24 presents reduced Hidden Size of 192, Num Heads of 4 and Depth of 6. This specific adjustment is motivated by the inherent nature shorter time series sequences, characterized by less intricate temporal dependencies and a comparatively smaller information manifold. Consequently, a more compact model—characterized by fewer attention heads and a smaller hidden dimension—is often sufficient to capture the underlying data distribution effectively without incurring unnecessary computational overhead or risking overfitting on a simpler generative task.

| Dataset Seq. Len. | N_FFT | Hop Length | Patch Size | Depth | Hidden Size | Num Heads | MLP Ratio |
|---|---|---|---|---|---|---|---|
| 24 | 11 | 3 | (2,1) | 6 | 192 | 4 | 4.0 |
| 64 | 31 | 11 | (4,1) | 8 | 384 | 6 | 4.0 |
| 128 | 63 | 15 | (8,1) | 8 | 384 | 6 | 4.0 |
| 256 | 127 | 32 | (16,1) | 8 | 384 | 6 | 4.0 |

Table 4: Hyperparameters for different sequence lengths.

# C ADDITIONAL VISUALIZATIONS

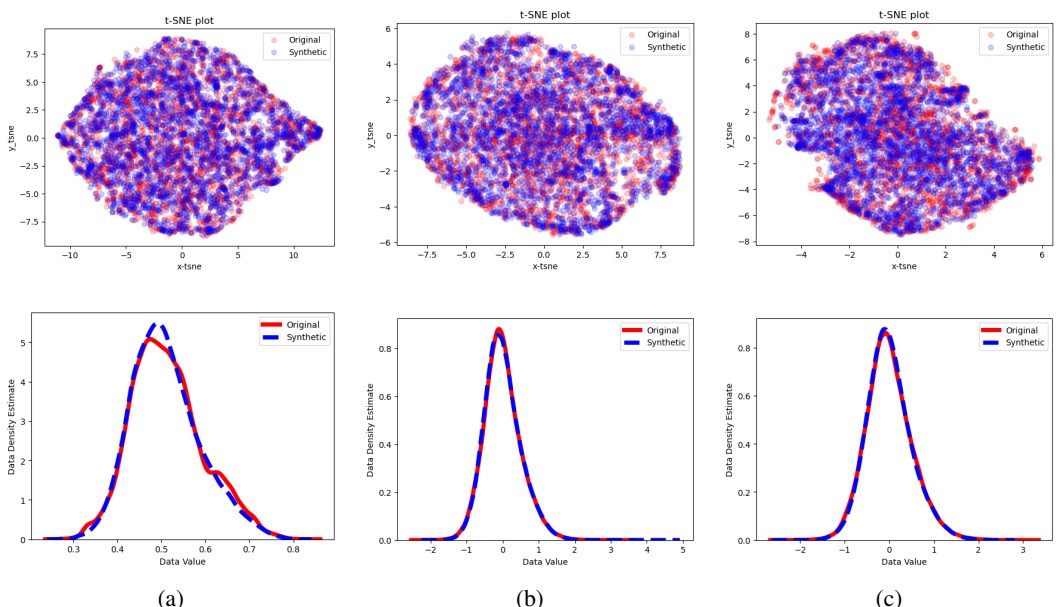

Figure 5: t-SNE (up) Kernel Density Estimation (bottom) of real vs generated samples of the **ETTh** dataset for sequence length 24(Fig. 5a), 64(Fig. 5b), 128(Fig. 5c). In red the original time series, in blue the generated one.

To complement the quantitative results and qualitative analyses presented in the main body, this section provides a more extensive set of visualizations the distributional alignment of generated samples for the ETTh dataset across multiple sequence lengths and a per-covariate breakdown of the temporal and spectral fidelity analysis.

Figure 5 provides additional qualitative validation for our model's performance on the ETTh dataset across varying sequence lengths (24, 64, and 128). The t-SNE plots (top row) demonstrate that the manifold of the generated samples (blue) consistently and comprehensively overlaps with that of the original data (red), indicating that ST-Diff successfully learns the global data structure. Furthermore, the Kernel Density Estimation (KDE) plots (bottom row) show a near-perfect alignment between the marginal distributions of the real and synthetic data, a consistency that is maintained even as the sequence length increases, underscoring the model's robustness and scalability.

Figure 6 and Figure 7 offer a comprehensive, per-covariate analysis of the temporal and spectral fidelity for the ETTh dataset (sequence length 24) and the fMRI dataset (sequence length 24), respectively. This complements the summarized results in the main paper. The figures systematically compare the Auto-Correlation Function (ACF) and Power Spectral Density (PSD) for the covariates in the datasets. The uniform consistency across all plots demonstrates that the high-fidelity generation is not an artifact of a few selected channels. Instead, ST-Diff robustly captures the unique temporal dependencies (ACF) and spectral characteristics (PSD) of each individual time series, providing strong evidence that our model learns the complete, multivariate data-generating process.

# D LARGE LANGUAGE MODEL USAGE

Large language models have been utilized to polish and refine writing for enhanced conceptual clarity, improving grammar, rephrasing sentences, and suggesting alternative word choices to make the text more concise and understandable.

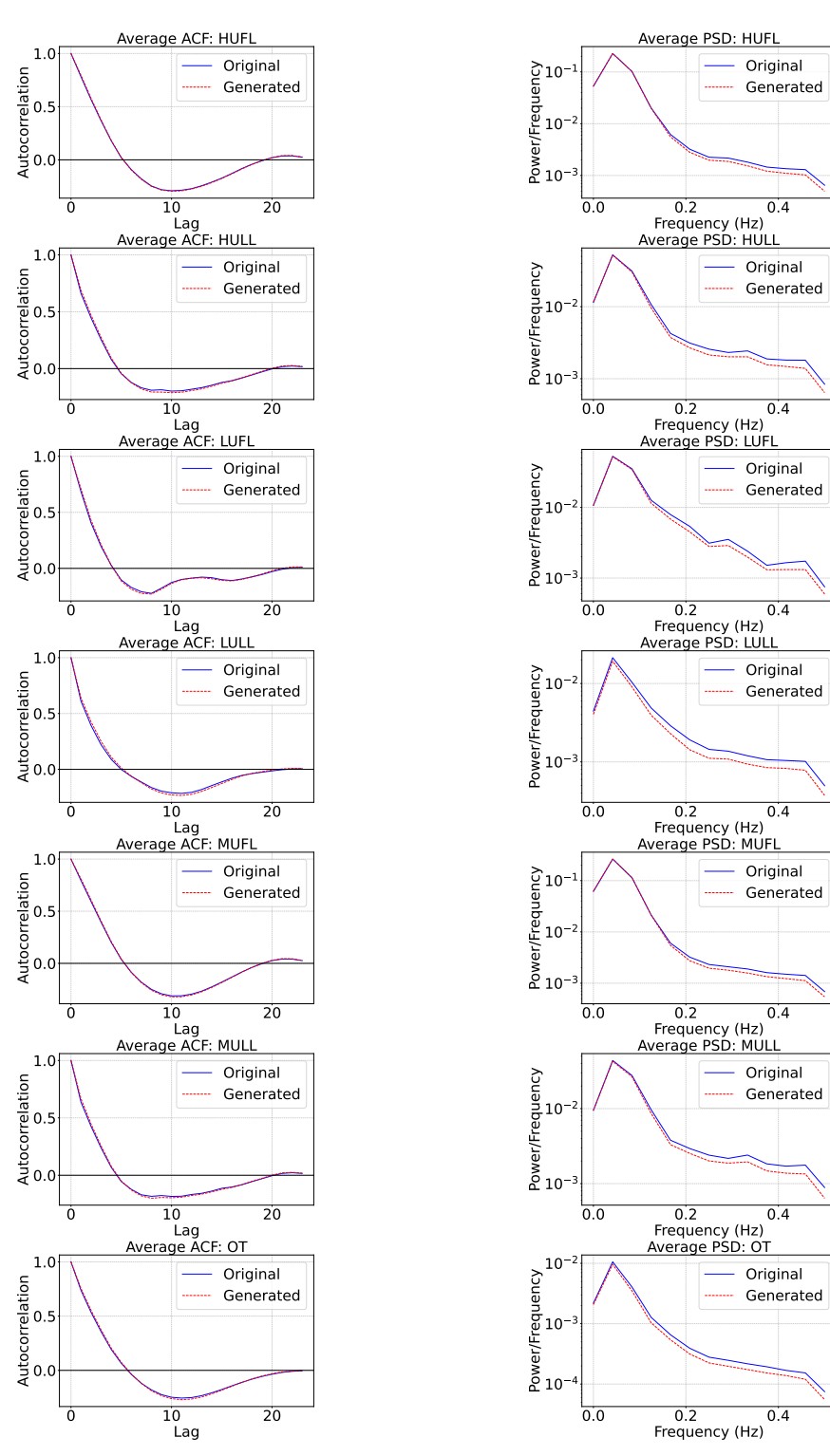

Figure 6: Complete analysis of Temporal and Spectral Fidelity on the ETTh dataset for seq_len 24. In the left column the Auto-Correlation Function (ACF), and in the right column the average Power Spectral Density (PSD) for real (blue, solid) and generated (red, dashed) samples across all the dataset covariates

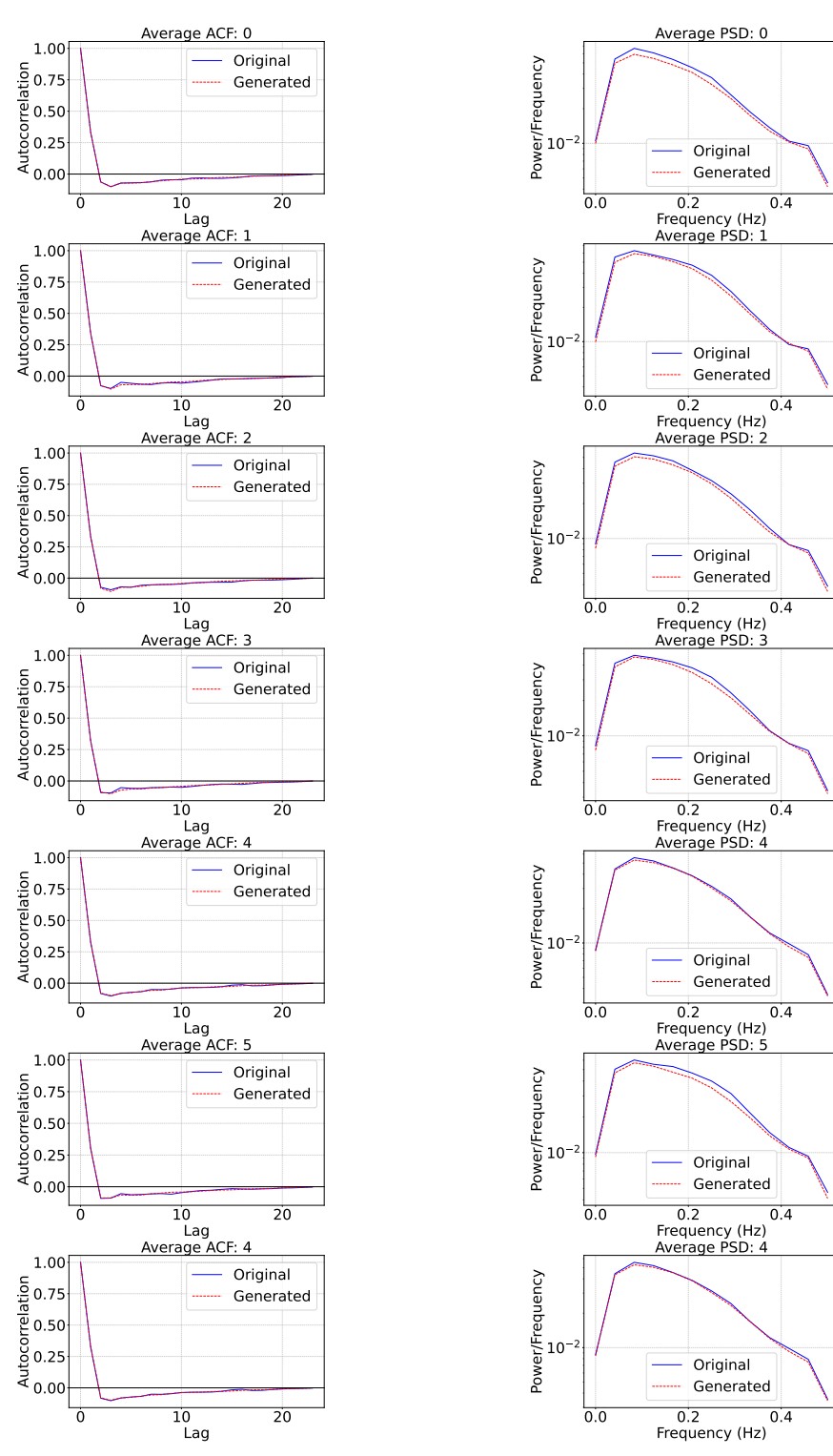

Figure 7: Complete analysis of Temporal and Spectral Fidelity on the fMRI dataset for seq_len 24. In the left column the Auto-Correlation Function (ACF), and in the right column the average Power Spectral Density (PSD) for real (blue, solid) and generated (red, dashed) samples across all the dataset covariates. Notice that just the first 7 covariates were provided, the other 43 are very similar and not included for simplicity.

