# OpenReview forum: "Time Series as Videos: Spectro-Temporal Generative Diffusion"
_ICLR.cc/2026/Conference — ICLR 2026 Conference Withdrawn Submission_

### Official Review · Reviewer_BV5s · 2025-10-24

**Soundness:** 2
**Presentation:** 3
**Contribution:** 3
**Rating:** 4
**Confidence:** 3

**Summary:**

This paper presents ST-Diff, a new generative framework for multivariate time series. The core idea is to transform time series into spectro-temporal videos using the Short-Time Fourier Transform (STFT) and then perform generation via a video diffusion model. The authors design a multi-axis Transformer architecture with frequency and covariate attention biases to model complex dependencies across temporal and spectral dimensions. Experiments are conducted on several datasets for unconditional generation, showing strong and consistent performance.

**Strengths:**

- The idea of treating time series as videos is creative and elegant, offering a fresh perspective on temporal generative modeling.
- The model design is well-motivated; leveraging STFT and spectral biases effectively captures periodicity and cross-channel relationships.
- Experiments demonstrate clear improvements over existing methods, validating the proposed approach for unconditional generation.

**Weaknesses:**

- All experiments are limited to unconditional generation, leaving the applicability to more practical conditional tasks (e.g., forecasting, imputation) unexplored.
- While STFT helps capture spectral structures, it changes the temporal representation and may weaken the model’s ability to learn direct temporal dependencies, thereby hindering effective conditional learning — an issue not discussed in the paper.
- Converting time series into videos introduces substantial computational and memory overhead, and the paper lacks systematic analysis of model efficiency or scalability.
- The architectural components (e.g., bias matrices) lack sufficient ablation studies to support their claimed contribution.

**Questions:**

See Weaknesses.

---

### Official Review · Reviewer_Mief · 2025-10-24

**Soundness:** 2
**Presentation:** 3
**Contribution:** 2
**Rating:** 4
**Confidence:** 4

**Summary:**

The paper introduces  a method that reframes multivariate time series as time–frequency “video” tensors using the STFT. By treating frequency and covariates as spatial dimensions, a custom video-diffusion architecture models the spectro-temporal dynamics. This approach yields state-of-the-art results in unconditional time-series generation and is proposed as a generalizable time-series-as-video paradigm. While a promising extension of **ImagenTime**, the work’s  empirical foundation is incomplete, requiring more comprehensive ablations, broader downstream results, and a thorough evaluation of computational trade-offs.

**Strengths:**

1. The core research question, leveraging advances in video diffusion for time-series generation, is timely and important for cross-modal progress in time-series generation.

2. Modeling trend and residual with distinct 3D channel pathways is interesting and potentially impactful. That said, additional comparisons (e.g., delay embeddings or simpler time-series→video mappings) would help quantify necessity versus sufficiency, which is currently lacking.

3. The manuscript is generally well written and easy to follow.

4. While the results are limited in breadth and should be expanded, the findings on specific tasks are convincing.

**Weaknesses:**

1. **Related work coverage.** Several relevant threads are omitted (e.g., [1], [2] and other recent time-series generative approaches). The survey should be broadened to better situate this contribution within the time-series generation literature.

2. **Limited evaluation relative to the closest baseline (*ImagenTime*):**
   - The paper claims improved handling of long sequences but does not compare on *ImagenTime*’s long-range benchmarks (1k–20k steps).
   - The paper claims broader applicability to diverse time-series tasks, yet does not report on tasks where *ImagenTime* already provides benchmarks (interpolation and extrapolation).

3. **Conceptual tension.** If time-series generation is treated as “video,” one would expect future video-model advances to transfer with little change. However, the method relies on architecture-specific adaptations (Section 4.3), raising questions about generality and future-proofing. Clarification is needed regarding how much modification future architectures would require.

4. **No complexity analysis.** Time and memory costs are not reported relative to contemporary SOTA baselines.

5. **Missing ablation on key component.** There is no ablation isolating the effect of the proposed component. Comparisons against simpler alternatives, such as duplicating the time series across channels, using delay embeddings, or other plausible baselines, would help clarify the added value of this design choice.

6. Minor typo: ``in order to handle non-stationarity taht is'' $\rightarrow$ ``in order to handle non-stationarity that is''.


[1] Generative modeling of regular and irregular time series data via koopman VAEs

[2] On the constrained time-series generation problem

**Questions:**

See weaknesses please.

---

### Official Review · Reviewer_uQmW · 2025-10-29

**Soundness:** 2
**Presentation:** 2
**Contribution:** 1
**Rating:** 2
**Confidence:** 5

**Summary:**

This paper proposes ST-Diff, an interesting paradigm for multivariate time series generation that reframes the task as video generation. The core idea is to use an STFT to transform the time series into a spectro-temporal "video" tensor, which is then modeled by a custom-designed video diffusion model. The architecture incorporates well-motivated inductive biases, such as factorized attention and learnable priors for covariate and frequency relationships.

**Strengths:**

- The core idea of treating a multivariate time series as a video (with dimensions Time, Frequency, Covariates) is intersting and intuitive. It provides a principled way to apply powerful and well-studied spatiotemporal architectures (video models) to the time series domain, explicitly preserving the temporal axis.

**Weaknesses:**

- The authors motivate their work by claiming that RNNs struggle with long-range dependencies and that image-based methods are not ideal. However, the only experiment that tests long sequences (Table 2, $L=64, 128, 256$ on a single domain) completely omits ImagenTime, which is the most related SOTA baseline and is specifically designed to handle long sequences. Without this direct comparison, the central claim that the "time-series-as-video" paradigm is superior for modeling long-term dependencies is unsubstantiated. This weakens the paper's core argument significantly.

- The model's performance on the "Sines" dataset is a major red flag. As shown in Table 1, ST-Diff's Predictive Score is 0.186, which is twice as high (worse) than all other reported baselines (TimeGAN, TimeVAE, and ImagenTime, all at 0.093). A model built on the Short-Time Fourier Transform should, in principle, excel at modeling a combination of sine waves. This fundamental failure on a simple, periodic task casts doubt on the model's robustness and the overall efficacy of the chosen representation. The authors provide no explanation for this poor result.

- The paper repeatedly argues that methods transforming sequences into static images (like ImagenTime) "collapse the temporal axis, precluding the use of models designed for spatiotemporal dynamics". This argument in not backed empirically. While the explicit time axis is rearranged, the temporal information is meticulously preserved spatially (e.g., via delay embedding or the STFT's time-frequency grid). These models do learn spatiotemporal dynamics, but with a 2D spatial inductive bias. The real (and unaddressed) question is why an explicit 3D spatiotemporal bias is superior to a 2D spatial one.

- The authors claim in the conclusion that ST-Diff "incurs higher computational and memory costs" than image- or time-based models. This is a critical trade-off that is never quantified. A video model operating on a 3D tensor is almost certainly more computationally expensive (in both training and inference) than an image model on a 2D tensor. A complexity analysis (e.g., FLOPs or memory vs. sequence length $L$) is necessary for a fair comparison, and its omission is a significant weakness.

- The authors introduce several new architectural components, most notably the anisotropic patching strategy and the learnable bias matrices ($B_C, B_F$). However, the paper provides no ablation studies to empirically justify these design decisions. While the rationale is intuitive (e.g., avoiding arbitrary spatial correlations among covariates), the paper would be significantly stronger if it demonstrated that these components provide a measurable benefit over simpler baselines (e.g., standard isotropic patching, no bias matrices or a model used for actual video generation).

- The literature review is incomplete. The authors miss relevant work on generative VAEs that also seek to model underlying dynamics, such as KoVAE (which uses Koopman operators) and other diffusion and transformer based (SDformer) methods [1,2,3,4].


[1] Naiman, Ilan, et al. "Generative modeling of regular and irregular time series data via Koopman VAEs." ICLR 2024.

[2] Jeon, Jinsung, et al. "GT-GAN: General purpose time series synthesis with generative adversarial networks." NeurIPS 2022

[3] Coletta, Andrea, et al. "On the constrained time-series generation problem." NeurIPS 2023.

[4] Zhicheng Chen, et al. "SDformer: Similarity-driven Discrete Transformer For Time Series Generation." NeurIPS 2024

**Questions:**

See Weaknesses.

---

### Official Review · Reviewer_JFhg · 2025-11-01

**Soundness:** 3
**Presentation:** 3
**Contribution:** 3
**Rating:** 4
**Confidence:** 2

**Summary:**

This paper proposes ST-Diff, a novel framework for unconditional multivariate time series generation that reframes time series as spectro-temporal videos using the Short-Time Fourier Transform (STFT). A custom spatio-temporal diffusion transformer is then trained to generate in this video-like domain, followed by inverse STFT to reconstruct time-domain samples. The authors argue that this “time-series-as-video” paradigm better captures joint temporal–spectral dynamics than existing time-domain or static-image approaches. Empirically, ST-Diff achieves state-of-the-art results across six diverse benchmarks and multiple evaluation metrics, with especially strong gains on high-dimensional, complex datasets like fMRI and Energy.

**Strengths:**

1. **Strong empirical performance**: ST-Diff significantly outperforms recent SOTA baselines (e.g., Diffusion-TS, ImagenTime) across 21 of 24 metric–dataset combinations, often by **order-of-magnitude margins**—particularly in distributional fidelity (Context-FID) and cross-covariate correlation modeling.
2. **Conceptually elegant and novel**: The core idea—treating time series as videos via STFT—is **well-motivated**, bridges signal processing and generative modeling.
3. **Comprehensive and principled evaluation**: The paper follows community best practices for unconditional generation, including:
    - Standard quantitative metrics (Discriminative, Predictive, Correlational, Context-FID),
    - Qualitative analyses (t-SNE, KDE, ACF, PSD),
    - Long-sequence scalability tests (up to L=256),
    - Per-covariate spectral/temporal fidelity checks (Appendix).

**Weaknesses:**

1. **Lacks ablation studies on core design choices**: Despite introducing multiple novel components (trend-residual decomposition, anisotropic patching, bias matrices, cross-covariance STFT loss), the paper provides **no ablation experiments** to isolate their individual contributions. This weakens the attribution of performance gains to the proposed “video” representation versus auxiliary modeling tricks.
2. **Missing comparison to a highly relevant baseline**: The paper omits **Crabbe et al. (ICML 2024)**, which performs diffusion in the **frequency domain**. While distinct from ST-Diff’s joint time–frequency approach, this baseline is directly relevant and its exclusion leaves a gap in evaluating whether explicit temporal evolution of spectra is truly necessary.
3. **Overstated novelty claim**: The assertion that this is “the first to bridge these domains” overlooks decades of **audio generation work** that treats spectrograms as images or videos (e.g., WaveNet + mel-spectrograms, DiffWave). The novelty lies in applying this to **general multivariate non-audio time series**, which should be clarified.
4. **No analysis of computational cost**: ST-Diff uses a spatio-temporal transformer on 3D tensors, which is **significantly more expensive** than time- or image-domain methods. Yet the paper provides **no data on training/inference time, memory usage, or parameter count**, limiting assessment of practical utility.
5. **Ambiguity in phase handling during iSTFT**: The paper assumes near-perfect invertibility but does not specify how **phase consistency** is ensured during generation (e.g., Griffin-Lim, trainable phase, or magnitude-only reconstruction). This is critical for time-domain fidelity.

**Questions:**

1. Could you provide **ablation results** for key components (e.g., removing trend decomposition, disabling bias matrices, using isotropic patching)? Which design choice contributes most to the performance gain?
2. How is **phase information** handled during iSTFT? Is the generated STFT complex-valued with consistent phase, or is phase estimated post-hoc?
3. Have you considered comparing against **Crabbe et al. (ICML 2024)**? Their frequency-domain diffusion model seems like a natural middle ground between time-domain and your spectro-temporal approach.

---

### Note · Authors · 2025-12-09

I have read and agree with the venue's withdrawal policy on behalf of myself and my co-authors.